# Impacts of Wildfire Smoke and Air Pollution on a Pediatric Population with Asthma: A Population-Based Study

**DOI:** 10.3390/ijerph20031937

**Published:** 2023-01-20

**Authors:** Linn E. Moore, Andre Oliveira, Raymond Zhang, Laleh Behjat, Anne Hicks

**Affiliations:** 1Department of Pediatrics, University of Alberta, Edmonton, AB T6G 1C9, Canada; 2Department of Electrical and Software Engineering, University of Calgary, 2500 University Drive, Calgary, AB T3L 2M6, Canada; 3Department of Computer Sciences, University of British Columbia, Vancouver, BC V6T 1Z4, Canada

**Keywords:** wildfire, wildland fires, forest fires, pediatric population, asthma, air pollution

## Abstract

Wildfires are increasing yearly in number and severity as a part of the evolving climate crisis. These fires are a significant source of air pollution, a common driver of flares in cardiorespiratory disease, including asthma, which is the most common chronic disease of childhood. Poorly controlled asthma leads to significant societal costs through morbidity, mortality, lost school and work time and healthcare utilization. This retrospective cohort study set in Calgary, Canada evaluates the relationship between asthma exacerbations during wildfire smoke events and equivalent low-pollution periods in a pediatric asthma population. Air pollution was based on daily average levels of PM_2.5_. Wildfire smoke events were determined by combining information from provincial databases and local monitors. Exposures were assumed using postal codes in the health record at the time of emergency department visits. Provincial claims data identified 27,501 asthma exacerbations in 57,375 children with asthma between 2010 to 2021. Wildfire smoke days demonstrated an increase in asthma exacerbations over the baseline (incidence rate ratio: 1.13; 95% CI: 1.02–1.24); this was not seen with air pollution in general. Increased rates of asthma exacerbations were also noted yearly in September. Asthma exacerbations were significantly decreased during periods of COVID-19 healthcare precautions.

## 1. Introduction

Asthma is the most common chronic respiratory disease in children, with significant costs associated with healthcare utilization [1,2], morbidity and mortality. Common triggers for asthma exacerbation include respiratory irritants. Air pollution is a known source of triggers for asthma exacerbation; in particular, small particulate matter (PM_2.5_) has been associated with respiratory morbidity, mortality and associated burdens to public health [3,4].

Outdoor air pollution contributes to morbidity and mortality, with an estimated three million premature deaths yearly attributable to air pollution exposures [5]. In the context of climate change, wildfires are becoming a growing source of air pollution secondary to their increase in size, intensity and the length of the annual fire season [6,7,8,9,10]. In Canada in 2019 alone, more than 1.8 million hectares were burned in nearly 4000 wildfires [10]. Healthcare utilization for acute cardiorespiratory events is typically increased during periods of elevated PM_2.5_ attributed to wildfire smoke exposure, which can be increased for days to weeks and is often observed for great distances from the original wildfire source [11,12,13,14,15,16]. 

While some reports suggest that children are at increased risk of acute respiratory events resulting in healthcare visits during periods of exposure to wildfire smoke [17,18,19,20], most reports on children’s health during wildfires are mainly associated with population-level data analysis focused on adult outcomes. While some data showed increases in asthma exacerbations, reported mainly from clinical billing data [14,21], it is not clear that all the patients had a confirmed asthma diagnosis [15,19,22]. In Alberta, Canada, about half of the estimated 12% of the population who have ever been diagnosed with asthma have poor baseline control [23]. Alberta is also a province that has been impacted by wildfires, both locally and by smoke from nearby provinces and the United States [24]. There is a dearth of information focused on the impact of wildfire smoke on pediatric populations [25]; while two existing cohort studies explored the impact of wildfire smoke [14,15], population-level evaluation has been through extracting pediatric data from predominantly adult studies [17]. To our knowledge, this is the first population-level analysis of the impact of wildfire smoke specifically on a pediatric population who have already met the criteria for a diagnosis of asthma. 

## 2. Materials and Methods

### 2.1. Study Setting

This study compared rates in asthma-related health visits associated with outdoor PM_2.5_ levels and attributed to wildfire smoke in a cohort of children aged 1–17 years with a diagnosis of asthma in Calgary, Alberta, Canada, between 1 January 2010 and 30 September 2021. Calgary is the largest metropolitan area in the province, growing from a population of approximately 1,122,129 to 1,372,178 over the 2010–2021 study period [26]. The most recent Statistics Canada Census data from 2016 reported 1,392,609 residents of whom 261,455 (18.8%) were 0 to 14 years old (51% male) [27]. The province of Alberta uses a single publicly funded healthcare system (Alberta Health Services) with universal health coverage for all residents. Provincial healthcare data can, thus, be assumed to reflect healthcare utilization at the population level [28].

### 2.2. Study Population and Associated Data

#### 2.2.1. Population and Data Linkage

This retrospective cohort study used linked data from inpatient and outpatient health visit records and pharmaceutical dispense data. Linkage between datasets was created based on unique personal healthcare identification numbers assigned to each residence of Alberta and recorded during all healthcare encounters. Information on socioeconomic status was derived from census data at the time of asthma diagnosis and linked to the cohort based on dissemination area codes. The cohort included all children living in the greater Calgary area with a diagnosis of asthma between the ages of one and 17 years old between January 2010 and September 2021, as identified from Alberta inpatient and outpatient billing codes using validated criteria [22]. Briefly, the diagnosis of asthma was defined as having at least two outpatient physician visits with an asthma diagnosis in the first diagnostic field in a two-year period, or at least one inpatient hospital visit with a diagnosis of asthma ever in any diagnostic field, coded by the International Classification of Diseases (ICD), ninth revision ICD-9-CM 493 or tenth revision ICD-10-CA J45–46 [29,30]. No exclusion criteria were applied. 

#### 2.2.2. Clinical and Pharmaceutical Data

Emergency department (ED) visits for asthma were obtained from ambulatory data according to an ICD-10 code starting with J45 in the first diagnostic field [29,30]. To ensure that ED visits associated with a singular asthma exacerbation were only counted once, exacerbations were defined as ED events that were at least 7 days apart from a previous exacerbation; multiple ED visits during a 7-day period were counted as a single event [15,21]. Severity of the asthma presentation during each ED visit was estimated using the 2016 version of the Canadian Triage and Acuity Scale (CTAS) score recorded at the time of the healthcare visit [31]. The CTAS score triages patient acuity as 1—resuscitation; 2—emergent; 3—urgent; 4—less urgent; 5—non-urgent. Additional codes, 0—deceased and 9—unconscious, were not included as the contribution of asthma was unclear. Information on hospitalizations for asthma was collected from the Discharge Abstract Database and defined as a recording starting with ICD-10 code J45 in the first diagnostic field. Admissions to the pediatric intensive care unit (PICU) were based on code 70 in any of the special care unit fields in hospitalization data. Information on medications prescribed for asthma was based on anatomical therapeutic chemical codes for reliever medications (short-acting beta-agonists), controller medications (including inhaled corticosteroids with or without long-acting beta-agonists), leukotriene receptor antagonists and oral corticosteroids dispensed within the first year following diagnosis.

### 2.3. Air Pollution Data

Air pollution data were obtained from the Government of Alberta “Access Air Quality and Deposition Data” website [32]. Hourly PM_2.5_ levels from multiple ground-level monitor sites within the Calgary region were summarized as 24 h averages and categorized using the 2020 Canadian Ambient Air Quality Standards (CAAQS) scoring system for PM_2.5_ (Table 1) [33].

PM_2.5_: particulate matter less than 2.5 µm in diameter.

Poor air quality events attributable to wildfires were identified by overlaying information from the Government of Alberta “Access Air Quality and Deposition Data” website [32], news reporting on wildfire smoke coverages and satellite images [34]. This method of identifying smoke exposures by using the combination of ground-level monitor data and satellite images has been used previously in Canadian analyses of healthcare utilization associated with wildfire smoke [35]. Dates with wildfire smoke were defined as dates with wildfire smoke in the area reported through various news outlets and confirmed by visual smoke coverage on satellite images and with a daily (24 h) average PM_2.5_ ≥ 20 µg/m^3^ (orange or red CAAQS zones). Days adjacent to confirmed wildfire smoke days with PM_2.5_ ≥ 20 µg/m^3^ were also considered as wildfire smoke days.

### 2.4. Statistical Analysis and Modeling

Data on cohort characteristics were summarized for each person and reported as count and percentages for binary variables and means and 95% confidence intervals (95% CI) for continuous variables. The number of ED visits was sorted based on the Calgary Region Airshed Zone data CAAQS PM_2.5_ zones during the study period, and daily rates were calculated by dividing the total number of ED visits per CAAQS PM_2.5_ zone by the number of days in each CAAQS PM_2.5_ zone. Poisson regression analysis was performed to assess the association between CAAQS PM_2.5_ zones, smoke days and rates of ED visits, and reported as incidence rate ratio (IRR) and 95% CI for each of the yellow, orange and red zones, and smoke days, while using the green CAAQS PM_2.5_ zone as the reference zone. Visual representations of the relationship between PM_2.5_ levels and rates of asthma exacerbations were created by obtaining a base image of the map of the greater Calgary region from Google maps and overlaying the points for asthma exacerbations via postal codes recorded at each healthcare encounter [36]. Graphs containing a three-day rolling average for both the number of asthma exacerbations and the air pollution index were created for each year. All statistical analyses were performed using Stata SE software version 17.0 (StataCorp LLC, College Station, TX, USA). Visual representative images of asthma exacerbations during low, high and smoke days, were created using the Pandas version 1.3.5 [37], Numpy version 1.21.6 [38,39] and matplotlib version 3.2.2 libraries, and graphs were created in Jupyter Core version 4.11.2 [40] and Python version 3.7.15 (Python Software Foundation, Beaverton, OR, USA) [41].

### 2.5. Ethics Statement

This study received approval from the University of Alberta Research Ethics Board (Pro00114647). Due to the retrospective and unidentifiable nature of the data, individual informed consent was not required. Data access and linkage were provided by the Alberta Health Services. Inquiries pertaining to the data or data access can be directed to research.administration@ahs.ca.

## 3. Results

### 3.1. Population Characteristics

A total of 57,375 children aged 1–17 years were diagnosed with asthma in the greater Calgary area between 1 January 2010 and 30 September 2021, based on either one hospitalization with asthma as a factor for the visit or two outpatient clinic visits with asthma as the primary reason for the visit [22]. As shown in Table 2, approximately 60% were male and the average age of asthma diagnosis was 7.31 years. Twenty-five percent of the children in the cohort had one or more ED visits for asthma, resulting in a total of 24,877 recorded ED visits for asthma over the study period. The average CTAS score for these ED visits was 2.67. Altogether, the mean number of ED visits per child was 0.43. A smaller proportion of children (3.7%) were hospitalized for asthma at any time during the study period, and 0.8 of children were admitted to PICU. Nearly half of all children (46%) were prescribed and filled a prescription for asthma reliever medications during the first year following diagnosis, while approximately one third of patients filled a prescription for asthma controller medications, and 5% of children had one or more dispenses for leukotriene receptor antagonists. Oral corticosteroids, an indicator of a severe asthma exacerbation [42], were dispensed to 6% of children in the cohort within the first year following the diagnosis of asthma. 

### 3.2. Air Pollution Characteristics

For most of the time (74%) between 1 January 2010 and 30 September 2021, the air quality in Calgary was good (green zone) according to the CAAQS air quality management level zones [33] for PM_2.5_, with 22% of days in the yellow zone and 2% of days displaying air quality in the orange and red zones each (Table 3). Of all days in the study period, there were a total of 60 days (1.4%) during which smoke from wildfires was present (Table 3).

### 3.3. Relationship between Air Pollution, Wildfire Smoke and Asthma Exacerbation

The average number of asthma exacerbations resulting in ED visits was 5.8 per day for the entire study period. The rate of ED visits per day ranged from 5.1 (orange zone days) to 6.5 (wildfire smoke days). Examples of asthma exacerbations based on location during different air quality periods are shown in Figure 1. While there was no significant difference in the risk ratio of asthma exacerbations resulting in ED visits on days with air quality in the yellow or red zone compared to the green zone, the risk for ED visits for asthma was lower on days when the air quality was in the orange zone (IRR: 0.88 [95% CI 0.80–0.96], Table 3).The risk of ED visits for asthma exacerbations on days with wildfire smoke was 1.13-fold (95% CI 1.02–1.24) compared to days with air quality in the green zone.

### 3.4. COVID-19 Public Health Precautions and Asthma Exacerbations

In Alberta, COVID lockdown extended from 15 March to 15 June 2020 [43]. Schools reopened in September 2020 and mask mandates were lifted 14 February 2021. Pediatric asthma visits to the ED were lower from March to August 2020, with a small concomitant increase in PM_2.5_ levels and asthma exacerbations in August and September 2020 (Figure 2). Asthma-related ED visits rose in March 2021, after school mask mandates were lifted, then increased significantly during the summer, coincident with PM_2.5_ (Figure 2). Asthma visits were minimal during COVID-19 pandemic precautions on “green zone” (good) air quality days (Figure 1).

**Figure 1 ijerph-20-01937-f001:**
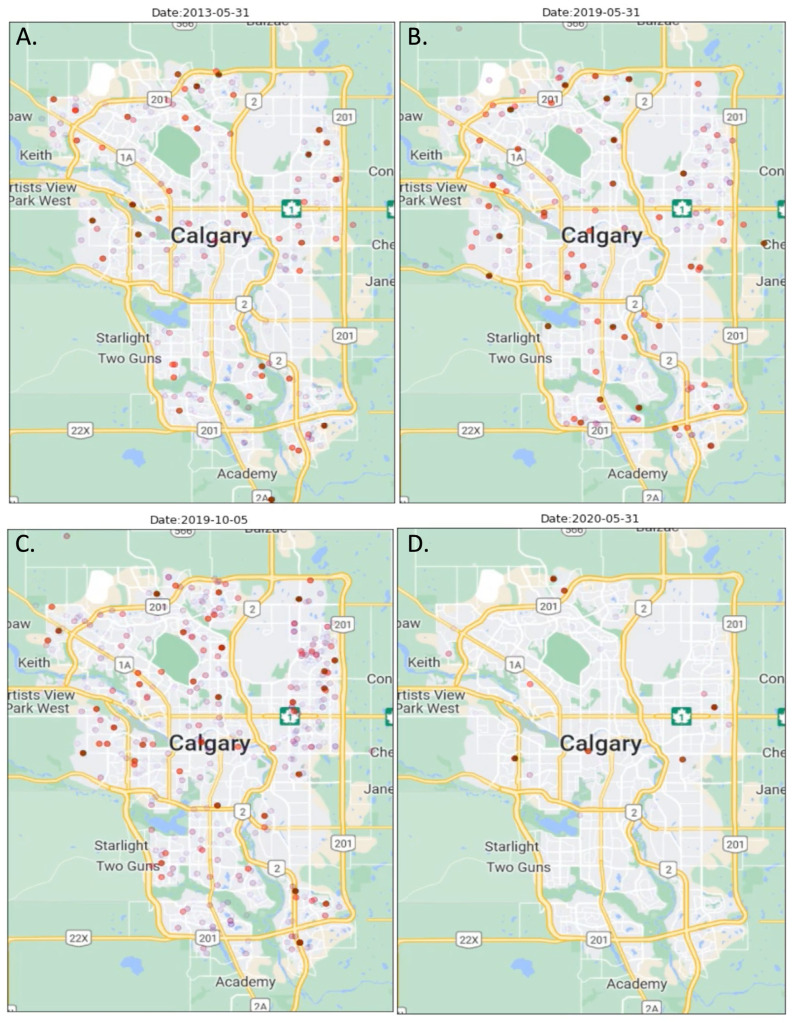
Locations of billing addresses on the date of emergency department visits for an asthma exacerbation for children aged 1–17 years. Asthma cases are indicated by red dots—the darkest dots are same-day cases, while faded dots indicate cases up to 3 days prior to and 3 days after the date indicated [36]. (**A**) 31 May 2013: a representative image of a day with no wildfire smoke present and “green zone” (good) air quality; (**B**) 31 May 2019: during a wildfire smoke event with elevated PM_2.5_ in the CAAQS “orange or red zone” for air quality [33]; (**C**) 10 September 2019: during a typical “September Spike” in asthma cases [44]; (**D**) 31 May 2020: during the COVID-19 pandemic lockdown period on a day with “green zone” (good) air quality [43].

**Figure 2 ijerph-20-01937-f002:**
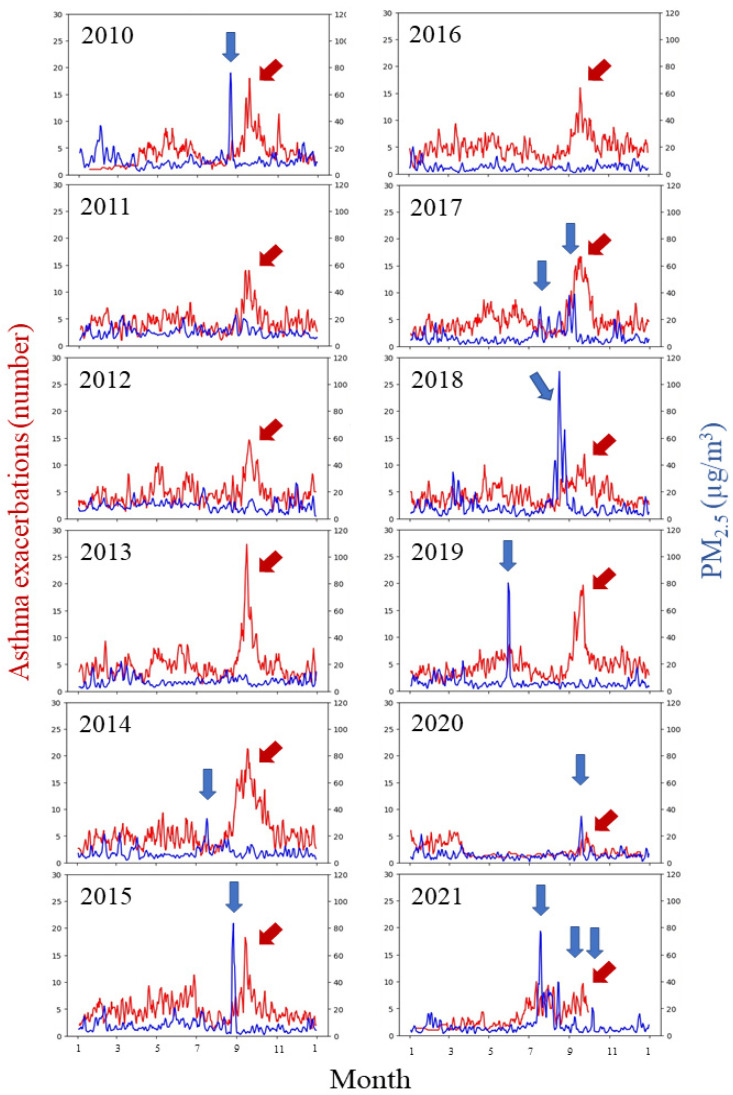
Three-day rolling averages of particulate matter (PM_2.5_) and cases of asthma exacerbation by year, 2010–2021. PM_2.5_ (hourly values reported from monitor sites in the Calgary region that were extracted from the Government of Alberta “Access Air Quality and Deposition Data” website [32] and summarized as 24 h averages) is shown in blue, with asthma exacerbations in red. The blue arrows indicate periods where wildfire smoke was present in Calgary [34]. Sources of wildfire smoke impacting Calgary were: Alberta (2010, 2014, 2017, 2019); British Columbia (2015, 2017, 2018, 2020, 2021); Saskatchewan (2021); United States (2015). The red arrows indicate the annual increase in asthma cases observed in the early fall, referred to as the “September-spike”.

## 4. Discussion

This retrospective population-level cohort study examined rates in pediatric asthma exacerbations in relation to air pollution and wildfires from January 2010 through September 2021 in Calgary, Alberta, Canada. Of the 57,375 children with asthma in the study, 14,120 (24.6%) had at least one ED visit for asthma and 2106 (3.7%) were hospitalized for asthma during the study period. While often coinciding with a spike in air pollution, the main driver for ED visits for asthma in Calgary during the study period appeared to be the return to school in the fall, resulting in the “September Spike” in asthma exacerbations [44]. Wildfire smoke days were recorded on 60 (1.4%) of the days and were associated with a 13% increase in ED rates for asthma. 

The city of Calgary, Canada, is a prairie city that is impacted regularly by smoke from wildfires and grass fires [45]. Secondary aerosol formation is responsible for about half of the air pollution in Calgary, while primary emissions come from traffic and industry, including upstream oil and gas activities, construction, agricultural activity and natural gas combustion—mainly for heating residential and commercial buildings [45,46,47]. Taking wind and other weather factors into account, increases in PM_2.5_ during winter can be attributed to local combustion, secondary aerosols and nearby industrial sites, whereas summertime elevations are mainly attributable to wildfires [45]. Asthma exacerbations (all ages, no specification of an asthma population) in Calgary have previously been shown to be increased in areas of poor air quality, where air pollution was attributable to traffic and industry, although the impact of air pollution on children and on people with a pre-existing asthma diagnosis was not evaluated [47]. In contrast, in our pediatric asthma cohort, moderately poor air quality (orange zone [33]) appeared protective; one possible explanation is that children with asthma avoid exposures during these periods, however this observation is unexpected and warrants further exploration. It is also plausible that there is a delayed onset of respiratory symptoms following exposure to high levels of PM_2.5_ [17].

Children are particularly vulnerable to wildfire smoke, with several studies noting increased symptoms, physician and emergency room visits and hospitalizations during periods of wildfire smoke [16,19,20,21,35,48]. This study focused on asthma exacerbations in a cohort of children with a pre-existing diagnosis of asthma. There are multiple other triggers, in addition to air pollution, for asthma in this population. Widespread viral infections in the first few weeks of school, in conjunction with fall aeroallergens, are thought to drive the “September spike” observed across North America [44]. This was clearly visible in 2010–2019 and 2021 in Calgary, with the exception being during the COVID-19 lockdown period in 2020 (Figure 2), and similar observations have been made in other regions [49]. This trend continued through the period where masking was required in schools, although somewhat attenuated. Asthma cases rose as soon as the lockdown was lifted, as of 14 April 2021. Periods of high levels of PM_2.5_ associated with wildfire smoke are also associated with increases in pediatric asthma presentations; this is somewhat obscured by high background noise, but was particularly evident in 2019, while in 2015 it was not possible to link wildfire smoke to asthma cases due to the overlap in the month of September. 

This study focused specifically on a pediatric asthma population, unlike others where the impacts of wildfire smoke on children have been reported as a part of population-level data analysis with an adult focus [16,19,20,21,35] or on an existing pediatric cohort, without characterization of underlying lung disease [14,15]. Limitations of the study include the lack of clinical detail available in the administrative databases, which rely on input and coding. Air quality monitoring is site-dependent and not necessarily reflective of air quality at the postal code level, and exposures to air quality events were assumed based on the postal code at the time of the of the healthcare claims, which leaves potential errors in the level of exposure. Medication purchase information was partial because not all dispensaries submitted records to the provincial tracking system and only prednisone, not dexamethasone, purchase information was available, limiting the validity of medication purchase results and exacerbations based on oral steroids [42]. In addition, during the study period, children in Calgary who visited the ED for asthma exacerbations were streamed into a community asthma education program designed to improve baseline control and decrease exacerbations, which may have impacted outcomes over time [50]. Wildfire smoke exposure was based on an air quality event reporting system, however some areas in Calgary are known to experience significant pollution events attributable to other sources that could act as confounders [47]. Importantly, seasonal variations in the rates of asthma exacerbations, due to both seasonal fluctuations in environmental allergens and the “September Spike” attributed to viral infections [44], may have impacted baseline asthma ED rates, thus decreasing the ability to separate the effect that was attributable to wildfire smoke-related air pollution. Other factors such as level of humidity, other emissions and pollutants, duration of exposure and temperatures may also play a role. Although COVID-19-related public health precautions [43] offered some insights into virus-driven asthma exacerbation trends, because schools remained open for much of the pandemic with minimal, if any, mitigations for airborne viral spread and limited adherence to strict masking policies, including high-density groupings with indoor mealtimes and poor technique, viral spread still occurred, as suggested by the September 2020 spike at the onset of the in-person fall term.

## 5. Conclusions

Asthma exacerbations are common in children, particularly those with a formal diagnosis of asthma. In the absence of pandemic lockdowns, there is a typical “spike” in September. During COVID lockdowns asthma exacerbations requiring ED presentation were minimal. The notable exception was during a wildfire smoke event, when asthma presentations to the ED increased in conjunction with poor air quality days. This study also showed that poor air quality due to wildfire was associated with a 13%increased risk of asthma exacerbations requiring ED visits. There was a less clear association with other air quality events, with moderately poor air quality from PM_2.5_ associated with fewer same-day asthma presentations. 

## Figures and Tables

**Table 1 ijerph-20-01937-t001:** Air quality categories (2020 Canadian Ambient Air Quality Standards) [33].

	Air Quality Management Level			
	Green	Yellow	Orange	Red
PM_2.5_ (µg/m^3^)	≤10	11–19	20–27	>27

**Table 2 ijerph-20-01937-t002:** Population characteristics for the cohort of children aged 1–17 years diagnosed with asthma [22] in Calgary, Alberta, between 2010–2021.

Characteristic	Number
Cohort, n total	57,375
Sex	
Male, n (%)	33,942 (59.2)
Female, n (%)	23,433 (40.8)
Age (years) at asthma diagnosis, mean (95% CI)	7.31 (7.27–7.36)
ED visits for asthma	
Total number of ED visits, n	24,877
Any—yes, n (%)	14,120 (24.6)
Any—no, n (%)	43,255 (75.4)
ED visits per patient, mean (95% CI)	0.43 (0.42–0.44)
CTAS score per ED visit, mean (95% CI)	2.67 (2.66–2.68)
Hospitalizations for asthma	
Total number of hospitalizations, n	2624
Any—yes, n (%)	2106 (3.7)
Any—no, n (%)	55,269 (96.3)
Number of hospitalizations for asthma per patient, mean (95% CI)	0.05 (0.04–0.05)
PICU admissions	
Total number of PICU admissions, n	566
Any—yes, n (%)	482 (0.8)
Any—no, n (%)	56,893 (99.2)
Number of PICU admissions for asthma per patient, mean (95% CI)	0.01 (0.01–0.01)
Medications in the first year after asthma diagnosis	
Short-acting beta-agonists, n (%)	26,344 (45.9)
Inhaled corticosteroids *, n (%)	18,632 (32.5)
Leukotriene receptor antagonists, n (%)	2748 (4.8)
Oral corticosteroids, n (%)	3687 (6.4)

ED: emergency department; CTAS: Canadian Triage and Acuity Scale; PICU: pediatric intensive care unit. * With or without long-acting beta-agonists.

**Table 3 ijerph-20-01937-t003:** ED visits divided by days in air quality zones (PM_2.5_) and during periods where air quality was impacted by wildfire smoke, in the greater Calgary region 2010–2021.

Air Quality Exposure Category	Days in Period, n (%)	ED Visits, n	ED Visits per Day, n (95% CI)	IRR (95% CI)
Green zone PM_2.5_< 10 µg/m^3^	3178 (74.1)	18,399	5.8 (5.6–5.9)	1 (reference category)
Yellow zone PM_2.5_ 10–19 µg/m^3^	938 (21.9)	5438	5.8 (5.5–6.1)	1.00 (0.97–1.03)
Orange zone PM_2.5_ 20–27 µg/m^3^	85 (2.0)	431	5.1 (4.2–5.9)	0.88 (0.80–0.96)
Red zone PM_2.5_ >27 µg/m^3^	90 (2.1)	501	5.6 (4.7–6.4)	0.96 (0.88–1.05)
Wildfire smoke days	60 (1.4)	391	6.5 (5.6–7.5)	1.13 (1.02–1.24)

## Data Availability

Data supporting the results can be obtained as follows: De-identified health record data can be obtained via contact with the study authors. Institutional ethics approval and a data sharing agreement with Alberta Health Services are necessary. Air quality data are publicly available from: https://www.google.com/url?q=https://www.alberta.ca/access-air-quality-and-deposition-data.aspx&sa=D&source=docs&ust=1670365795993690&usg=AOvVaw2R7Wf_hlPOsL_AIhjAlnM7 accessed on 10 November 2020; wildfire smoke data are publicly available from: https://weather.gc.ca/warnings/weathersummaries_e.html, accessed on 10 November 2020.

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
