# Peer review of "Impacts of Wildfire Smoke and Air Pollution on a Pediatric Population with Asthma: A Population-Based Study"

_ijerph, 2023, doi:10.3390/ijerph20031937_

Round 1
Reviewer 1 Report
To the authors,
please, see the attached file.

Author Response
Thank you for your comments. We feel that in addressing these we have had the opportunity to significantly improve our manuscript. Please see our response to each of the comments addressed below.
The manuscript entitled ‘’Impacts of wildfire smoke and air pollution on a pediatric population with asthma: a population-based study’’ deals with the impact of wildfire smoke on a cohort of children aged 1-17 years with a diagnosis of asthma in Calgary from 2010 to 2021. I like conciseness. However, the authors did not clearly discuss information relevant to environmental epidemiology such as exposure in terms of concentration, time extent, and representativeness of the measured or modelled values for the receptors.
2.3. Air pollution data
It is not clear how the authors evaluated exposure to ambient PM2.5 deriving from wildfire events. The authors should better describe the following:
- For each wildfire event, how did the authors define the ground monitoring stations pertinent to the adverse health outcomes? The authors mentioned the work by Henderson et al. 2011. Henderson et al. (2011) evaluated the representativeness of ground monitoring stations with respect to exposure (smoke event and receptors) by means of modelling (i.e., ArcGIS and CALPUFF model) as well as a method for assessing the area covered by smoke.
Response: Since we evaluated health outcome events occurring in a relatively small and well-defined area (Calgary greater region) the exposure to PM2.5 due to wildfire smoke was considered to be equal for all receptors. As such, we did not use modelling for assessing the area covered by smoke. Days of wildfire smoke affecting the city of Calgary were identified via information from satellite images available via worldview.earthdata.nasa.gov, news reports of smoke hazards in Calgary linked to confirmed wildfires and were ultimately confirmed if the PM2.5 levels were above ≥ 20 μg/m3. The reference to Henderson et al (2011) refer to to the two-step approach of using both air quality monitors and satellite images to confirm smoke events. Based on the reviewer’s question, this has been clarified in the revised version of the manuscript on page 3, line 118-127 as follows:
“Poor air quality events attributable to wildfires were identified by overlaying information from the Government of Alberta “Access Air Quality and Deposition Data” website32, news reporting on wildfire smoke coverage, and satellite images34. This means of identifying smoke exposures by using the combination of ground-level monitor data and satellite images has been used previously in Canadian analyses of healthcare utilization associated with wildfire smoke48. Dates with wildfire smoke were defined as dates with wildfire smoke in the area reported through various news outlets and confirmed by visual smoke coverage on satellite images and with a daily (24hrs) average PM2.5 ≥ 20 µg/m3 (orange or red CAAQS zones). Days adjacent to confirmed wildfire smoke days with PM2.5 ≥ 20 µg/m3 were also considered as wildfire smoke days.”
- PM2.5 is generated via both direct emission and secondary formation. How did the authors evaluate the entity of impact of wildfire smoke on air quality? Specifically, what were the base values of ambient PM2.5 before the smoke events?
Response: We agree with the reviewer that PM2.5 can be come from numerous sources including transportation and industry and is not exclusively seen following wildfires; however, in the city of Calgary, smoke from wildfires is the predominant source of high levels of PM2.5 during summer months (Bari et al, 2018). As this is a prominent and important topic in relation to the theme of the current study, the second paragraph of the discussion has been dedicated to the topic of primary versus secondary emissions in Calgary (page 9, line 241-256). As explained in the response to comment #1, days of wildfire smoke affecting the city of Calgary were identified via information from satellite images available via worldview.earthdata.nasa.gov, news reports of smoke hazards in Calgary linked to confirmed wildfires, and if the PM2.5 levels were above ≥ 20 μg/m3. Dates adjacent to confirmed wildfire smoke days were also considered to be impacted of wildfire smoke if the PM2.5 levels remained above ≥ 20 μg/m3 due to the high likelihood of the smoke coming from the same source despite the absence of additional news reports. The average daily levels of PM2.5 recoded during the period of the study not including the days identified as affected by wildfire smoke was 7.87 μg/m3. High PM2.5 levels during winter months were not included in the analysis as a) they could not be linked to concurrent wildfire events with smoke patterns covering the Calgary region, and b) could be due to temperature inversions resulting in fluctuating air quality which are known to affect the area where Calgary is located during winter months.
- In environmental epidemiology it is too vague to define exposure to a pollution event with no defined concentration values (i.e., PM2.5 ≥ 20 μg/m3 ) and time extent (i.e., adjacent days).
Response: Please see the response to comment number 2 above.
Line 120: …(Henderson). Please, revise reference format.
Response: This has been updated in the revised version of the manuscript (page 3, line 123).
Lines 173-175. ‘’ There were 64 days (1.5%) distributed through 2010, 2014, 2015, and 2017-2021 during which smoke from wildfires was present for at least one day ‘’ It is not clear. There were 64 days during which smoke was present for at least one day.
Response: based on the reviewer’s suggestion, the sentence regarding presence of wildfire smoke has been revised as follows (page 4, line 178-179):
“Of all days in the study period, there were a total of 60 days (1.4%) during which smoke from wildfires was present (Table 3).”
Figure 1
- Please, check the unit of y axis for the PM2.5 graph. The usual unit for PM2.5 is µg/m3 not mg/m3.
Response: This has been updated in the revised version of the manuscript (page 7, Figure 1).
- Panel 2013, please check the upper limit of y axis of asthma exacerbations because it looks like the maximum values are not shown.
Response: This has been updated in the revised version of the manuscript (page 7, Figure 1).
- Please, specify if the reported PM2.5 values are measured or modelled. If measured, specify what type of monitoring station/stations (e.g., urban background). What type of values (e.g., daily average) are shown? For a city of more than 1 million people, it is not representative of air quality levels to consider just one measurement.
Response: Based on the reviewer’s feedback, to establish a more accurate measure of PM2.5 levels we have incorporated the data from numerous measuring sites in the greater Calgary region in the updated analysis. In total, there were six measuring sites recording hourly data over the 12 years of the study. Using multiple sites to calculate the average hourly PM2.5 across the city, we have in the updated analysis used these numbers to calculate the 24-hour average PM2.5-levels. This resulted in a small change in the distribution of the number of days within each air quality zones (as defined by the 2020 Canadian Ambient Air Quality Standards) and the incidence risk ratio for ED visits during wildfire smoke days went from IRR: 1.10 (95% CI: 0.99 – 1.21) to IRR: 1.13 (95% CI: 1.02 – 1.24). While the IRR is now statistically significant, the clinical significance remains small. The methods pertaining to air quality data (page 3, line 109-127), the results in table 3 (page 9, Table 3), the result section (page 4-5, line 173-190), discussion, and conclusion have all been updated to reflect these changes in the revised version of the manuscript.
Additionally, we have further clarified how the PM2.5 data was obtained in the figure legend for figure 1 in the revised version of the manuscript (page 6-7, line 207-209) as follows:
“PM2.5 (hourly values reported from numerous monitor sites in the Calgary region were extracted from the Government of Alberta “Access Air Quality and Deposition Data” website32 and summarized as 24h averages) is shown in blue,…”
- Looking at the 2010, 2014, 2019 panels, it is quite evident that the spikes in PM2.5 levels are not associated with spikes in the number of asthma exacerbations.
Response: We agree with the reviewer that the increased levels of PM2.5 that seen concurrently with reported wildfire smoke coverage in Calgary in 2010, 2014 and 2019 do not appear to be linked to an increase in asthma exacerbations. To further explain that the arrows are not indicative of a cause and effect-relationship between wildfire smoke and asthma exacerbations, it has been clarified that the red arrows are showing the “September spike” (rise in number of asthma exacerbations seen annually in the fall) in the revised version of the manuscript (Page 7, line 213-214).
Figure 2.
- ‘’a’’ and ‘’b’’ labels are missing on the respective panels.
Response: The revised figure 2 with A and B labels have been added to the manuscript in place of the one missing the A and B labels.
- The difference between ‘’a’’ and ‘’b’’ panels is not so readily evident.
Response: The difference between the panels may not appear evident as the average number of ED events between days with low PEM2.5 (green zone) and days affected by wildfire smoke were 5.8 and 6.5, respectively. The images in Figure 2 are representative of typical days for each of the four situations of interest (low air pollution, wildfire smoke + high air pollution, the September spike in ED cases, and during the COVID-lockdown). Despite the images not displaying great differences, we hope the reviewer agrees with the importance of showcasing images representative of each of these situations.
- ‘’May 31, 2013, a typical year ‘’ It is shown just a day but the comment refers to a year. Please, revise.
Response: Based on the reviewer’s feedback, we have revised the legend to “May 31, 2013: a representative image of a day with no wildfire smoke present and “green zone” (good) air quality”. Page 7, line 220.
Table 3.
- ‘’Orange zone PM2.5 20-17 μg/m3 ‘’ Please, revise the range of concentrations.
Response: This has been updated to 20-27 μg/m3 in the revised version of the manuscript, page 8, Table 3.
- ‘’Days in period ‘’ Please, specify the period (maybe in the title of the table or in a table footnote).
Response: The timeframe for the study (in the greater Calgary-region 2010-2021) has been specified in the table heading for Table 3 in the revised version of the manuscript, page 8, line 228.
- ‘’1 (ref.) ‘’ Please, revise.
Response: The rates in ED events during days of “green zone” PM2.5-values was used as the reference value for calculating the IRR, as indicated in the methods. To clarify this, “ref” has been spelled out (reference category) in the revised version of the manuscript, page 8, Table 3.
- ‘’Forest fire smoke days ‘’ Is a reference missing?
Response: “Forest fire smoke days” the is label of one of the categories, similar to the ones listed above, and has been defined and referenced in the methods section. To clarify that the left column of Table 3 is displaying categories/groupings, the word “category” has been added to the top row (page 8, Table 3).
What is the relation between Figure 1 and Table 3? Table 3 reports 64 forest fire smoke days. However, figure 1 shows a few peaks in PM2.5 concentration.
Response: While Table 3 reports on daily numbers and rates, but as indicated in the methods section the data in Figure 1 is displayed as the rolling average (the text has been corrected to three-day rolling average) and include more than one day. This information has been added to the figure legend for Figure 1 in the revised version of the manuscript (page 6, line 206).
3 3.3. Relationship between air pollution and asthma exacerbation
Please, avoid confusion between air pollution and days with wildfire smoke. Moreover, ED visits per day are higher when air quality is denoted as green and yellow zones compared to orange and red zones. The authors should discuss why ED visits are higher during days with PM2.5 concentration < 19 µg/m3.
Response: Based on the reviewer’s comment, the subheading has been revised to also included “wildfire smoke” to not confuse it with other air pollution. The higher rates of ED visits during days of low PM2.5 was an unexpected finding and we believe it could be due to an avoidance of exposure during days of high PM2.5 in the pediatric asthma population we studied. It is also possible that there is a delayed effect from high PM2.5 levels not seen on the days immediately affected by high levels of PM2.5. These possible explanations have been added to the discussion section of the manuscript (page 8, line 251-255).
Lines 282-285.
This sentence is not clear. Please, rephrase this sentence.
Response: We thank the reviewer for pointing this out and have reworded the sentence as follows (page 9, line 290-294):
“Importantly, seasonal variations in the rates of asthma exacerbations due to both seasonal fluctuations in environmental allergens and the “September Spike” attributed to viral infections43 may have impacted baseline asthma ED rates and thus decreasing the ability to separate the effect attributed to wildfire smoke-related air pollution.”
Reviewer 2 Report
Dear Editor In-chief
The manuscript is about relationship between the air pollution due to the wildfires and the asthma exacerbation in children.
- First it should be said that the relationship between the asthma exacerbation and PM or other air pollutants resulted by the wildfire is not a new point. so I expect to see some new point of view such as the load of the air pollutants and the increased case numbers of the asthma exacerbation. Or a time series between them.
-Seeking for find the strong relationship between the kind of pollutants from the wildfire, duration of the exposure and influence of other factors such as the relative humidity could be interesting.
- It can be more discussed about the effect of covide-19 period and the asthma. How this can be related to the air pollution?
kind regards,
Author Response
Thank you for your comments. We appreciate your feedback and hope that in addressing these concerns we have substantially improved our manuscript. Please see our response addressing each comment directly below.
The manuscript is about relationship between the air pollution due to the wildfires and the asthma exacerbation in children.
- First it should be said that the relationship between the asthma exacerbation and PM or other air pollutants resulted by the wildfire is not a new point. so I expect to see some new point of view such as the load of the air pollutants and the increased case numbers of the asthma exacerbation. Or a time series between them.
Response: We thank the reviewer for these suggestions. The purpose of this study was to give a descriptive overview of the impact of PM2.5 and wildfire smoke on asthma exacerbations in a population-level pediatric cohort with confirmed asthma, as a recent systematic review (Henry et al 2021) identified reports on pediatric respiratory outcomes in children with asthma following wildfires as inadequate. To provide some insight to the degree children with asthma are affected by poor air quality and wildfire smoke, we conducted statistical analysis on the rates of ED visits during on days within the different PM2.5 zones, as suggested by 2020 Canadian Ambient Air Quality Standards, and during wildfire smoke days. As displayed in table 3, children with asthma do not seem to be suffering from increased rates of asthma exacerbations (using ED visits for asthma as a proxy) when the air quality is bad (orange or red zones) which might be indicative of reduced exposure to the outdoor environment during these days. It could also be indicative of a delayed onset of asthma-related symptoms. This has been further discussed on page 8, line 251-255 of the revised manuscript. We did however find a small but significant increase in the risk for ED visits among children with asthma during forest fires (IRR: 1.13; 95% CI: 1.02-1.24) compared to days with low air pollution. We hope the reviewer agrees with this approach of this report describing the responses to wildfire smoke in a well-defined pediatric asthma population.
-Seeking for find the strong relationship between the kind of pollutants from the wildfire, duration of the exposure and influence of other factors such as the relative humidity could be interesting.
Response: We agree with the reviewer that the question regarding duration of exposure as well as other influences, including the level of humidity, are important to take into consideration. In addition, some previous studies have reported on delayed symptoms/exacerbations/healthcare utilization following exposure to poor air quality and wildfire smoke. While this is out of the scope for the current descriptive study, we have added a section on this in the discussion of the revised manuscript to further highlight the need for future studies in this field addressing these important topics (page 9, line 294-295).
- It can be more discussed about the effect of covide-19 period and the asthma. How this can be related to the air pollution?
Response: The COVID-19 pandemic occurred towards the end of the study period and a significant reduction in asthma-related ED visits can be seen during government-mandated restrictions in social events and requirements to wear masks on the graphs in Figure 1. To highlight this, we have also included a representative image of Calgary during the COVID-19 lockdown displaying a significantly reduced number of ED visits during this time compared to any other timepoint. The impact of the COVID-19 pandemic is also discussed on page 9, line 263-271 of the discussion, as well as in the abstract and the conclusion for the study.
Reviewer 3 Report
The topic of the manuscript is very important as it highlights how wildfires are a significant source of air pollution and the key role they play in increasing cardiorespiratory diseases such as asthma. Considering asthma as the most common chronic disease of childhood, the text illustrates the significant social costs in terms of morbidity, mortality, loss of school and work time, and health care use to which poorly controlled asthma can lead. Based on a cohort study, the text provides an overview of the relationship between asthma exacerbations during fire smoking events and equivalent periods of low pollution in an asthmatic pediatric population. The results showed that fire-smoking days demonstrate an upward trend in asthma exacerbations compared to the baseline period and this was not found with air pollution in general. Furthermore, a decrease in asthma exacerbations was observed during COVID-19 health precaution periods. After making the suggested changes (listed below), this article may be accepted for publication in the International Journal of Environmental Research Public Health.
Suggested modifications:
Line 3: In the title of the manuscript, replace “a population-based study” with “A population-based study”.
Line 13: replace “including asthma. Asthma is the…” with “including asthma which is the…”.
Line 25: perhaps it would be better to replace the keyword “pediatrics” with “pediatric population”.
Line 63: replace “Sept 30” with “September 30”.
Line 114: insert a dot after the term “(2020 Canadian Ambient Air Quality Standards)”.
Line 115: insert a dot after the term “diameter”.
Line 195: insert a dot after the term “Figure 2”.
In Figure 2, in the first image, enter A. in the top left-hand corner, and in the second image, also in the top left-hand corner, enter B.
Line 214: after the word air quality replace the dot with a semicolon.
Line 215: after the word air quality replace the dot with a semicolon.
Line 216: after the word asthma cases replace the dot with a semicolon.
In Table 3 replace commas by a hyphen. Example: replace (5.7, 5.9) by (5.7- 5.9).
Line 244: The hypothesis of why poor air quality is protective towards pediatric asthma cohort is weak . If possible, try to provide a more exhaustive explanation.
Author Response
Thank you for your comments. In addressing these, we have improved our manuscript. Please see our individual responses to your recommendations below.
The topic of the manuscript is very important as it highlights how wildfires are a significant source of air pollution and the key role they play in increasing cardiorespiratory diseases such as asthma. Considering asthma as the most common chronic disease of childhood, the text illustrates the significant social costs in terms of morbidity, mortality, loss of school and work time, and health care use to which poorly controlled asthma can lead. Based on a cohort study, the text provides an overview of the relationship between asthma exacerbations during fire smoking events and equivalent periods of low pollution in an asthmatic pediatric population. The results showed that fire-smoking days demonstrate an upward trend in asthma exacerbations compared to the baseline period and this was not found with air pollution in general. Furthermore, a decrease in asthma exacerbations was observed during COVID-19 health precaution periods. After making the suggested changes (listed below), this article may be accepted for publication in the International Journal of Environmental Research Public Health.
Suggested modifications:
Line 3: In the title of the manuscript, replace “a population-based study” with “A population-based study”.
Response: This has been updated in the revised version of the manuscript (page 1, line 3).
Line 13: replace “including asthma. Asthma is the…” with “including asthma which is the…”.
Response: This has been updated in the revised version of the manuscript (page 1, line 13).
Line 25: perhaps it would be better to replace the keyword “pediatrics” with “pediatric population”.
Response: This has been updated in the revised version of the manuscript (page 1, line 25).
Line 63: replace “Sept 30” with “September 30”.
Response: This has been updated in the revised version of the manuscript (page 2, line 64).
Line 114: insert a dot after the term “(2020 Canadian Ambient Air Quality Standards)”.
Response: This has been updated in the revised version of the manuscript (page 3, line 115).
Line 115: insert a dot after the term “diameter”.
Response: This has been updated in the revised version of the manuscript (page 3, line 116).
Line 195: insert a dot after the term “Figure 2”.
Response: This has been updated in the revised version of the manuscript (page 5, line 199).
In Figure 2, in the first image, enter A. in the top left-hand corner, and in the second image, also in the top left-hand corner, enter B.
Response: The revised figure 2 with A and B labels have been added to the manuscript in place of the one missing the A and B labels.
Line 214: after the word air quality replace the dot with a semicolon.
Response: This has been updated in the revised version of the manuscript (page 8, line 221).
Line 215: after the word air quality replace the dot with a semicolon.
Response: This has been updated in the revised version of the manuscript (page 8, line 223).
Line 216: after the word asthma cases replace the dot with a semicolon.
Response: This has been updated in the revised version of the manuscript (page 8, line 224).
In Table 3 replace commas by a hyphen. Example: replace (5.7, 5.9) by (5.7- 5.9).
Response: This has been updated throughout Table 3 in the revised version of the manuscript (page 8).
Line 244: The hypothesis of why poor air quality is protective towards pediatric asthma cohort is weak . If possible, try to provide a more exhaustive explanation.
Response: This was an unexpected finding and while any attempt at explaining this phenomenon is highly speculative, the issues of delayed onset respiratory symptoms and avoidance of exposure have both been highlighted as possible reasons on page 8, line 251-255 of the discussion.
Round 2
Reviewer 1 Report
To the authors,
The authors answered to the comments I made on the manuscript. The manuscript was revised by the authors following the comments and suggestions I made.
Before publication, I suggest increasing quality of Figure 2.
Author Response
Thank you for your comments, which we believe have enabled us to improve our submission.
Before publication, I suggest increasing quality of Figure 2.
As suggested, we have maximized the quality of Figure 2 in the updated manuscript.
Reviewer 2 Report
Thank you for the revisions. I suggest the finding about the asthma and Covid-19 should be bolded in the abstract and results.
Regards,
Author Response
Thank you for your comments, which we believe have enabled us to improve our submission.
Thank you for the revisions. I suggest the finding about the asthma and Covid-19 should be bolded in the abstract and results.
In response to this suggestion, we have bolded the findings about asthma and COVID-19 in the abstract, results, and discussion.